# Plasmon-driven nanowire actuators for on-chip manipulation

Shuangyi Linghu [1,7], Zhaoqi Gu [1,7], Jinsheng Lu [2], Wei Fang [2], Zongyin Yang [3], Huakang Yu[4], Zhiyuan Li[4], Runlin Zhu [1], Jian Peng [1], Qiwen Zhan [1,5], Songlin Zhuang[1], Min Gu [6] & Fuxing Gu [1✉]

Chemically synthesized metal nanowires are promising building blocks for next-generation photonic integrated circuits, but technological implementation in monolithic integration will be severely hampered by the lack of controllable and precise manipulation approaches, due to the strong adhesion of nanowires to substrates in non-liquid environments. Here, we demonstrate this obstacle can be removed by our proposed earthworm-like peristaltic crawling motion mechanism, based on the synergistic expansion, friction, and contraction in plasmon-driven metal nanowires in non-liquid environments. The evanescently excited surface plasmon greatly enhances the heating effect in metal nanowires, thereby generating surface acoustic waves to drive the nanowires crawling along silica microfibres. Advantages include sub-nanometer positioning accuracy, low actuation power, and self-parallel parking. We further demonstrate on-chip manipulations including transporting, positioning, orientation, and sorting, with on-situ operation, high selectivity, and great versatility. Our work paves the way to realize full co-integration of various functionalized photonic components on single chips.

[1] Laboratory of Integrated Opto-Mechanics and Electronics, Shanghai Key Laboratory of Modern Optical System, Engineering Research Center of Optical Instrument and System (Ministry of Education), University of Shanghai for Science and Technology, 200093 Shanghai, China. [2] State Key Laboratory of Modern Optical Instrumentation, College of Optical Science and Engineering, Zhejiang University, 310027 Hangzhou, China. [3] Cambridge Graphene Centre, University of Cambridge, Cambridge CB3 0FA, UK. [4] School of Physics and Optoelectronics, South China University of Technology, 510641 Guangzhou, China. [5] Department of Electro-Optics and Photonics, University of Dayton, 300 College Park, Dayton, OH 45469, USA. [6] Centre for Artificial-Intelligence Nanophotonics, School of Optical-Electrical and Computer Engineering, University of Shanghai for Science and Technology, 200093 Shanghai, China. [7] These authors contributed equally: Shuangyi Linghu, Zhaoqi Gu. ✉email: gufuxing@usst.edu.cn

All-photonic integrated circuits have the potential to meet the next-generation information technique demands of low power consumption and high-speed processing. Because of the excellent properties of high crystallinity and smooth surface, chemically synthesized freestanding metal nanowires have emerged as the most promising building blocks for photonic circuits[1,2]. Due to the high loss of surface plasmon polaritons (SPPs), metal nanowires need to be integrated with low-loss dielectric interconnects and waveguides such as silica micro/nanofibres and semiconductor nanowires, to construct hybrid photonic-plasmonic circuits and systems[3–8]. However, despite extensive advances in research, the lack of efficient methods to manipulate single metal nanowires (such as moving, positioning, and sorting) with high-precision controllability and versatility severely impedes co-integration of hybrid photonic and plasmonic components on a chip. The main cause is small size-induced strong adhesive forces of metal nanowires to substrates in non-liquid environments (such as van der Waals forces and electrostatic forces)[9,10]. Typically, the adhesion of micro/nanoscale objects to substrates in air environments can reach a level of ~μN, which greatly exceeds the typical value of the exerting force (~pN) on matters by light momentum. Thus the commonly used optical force-based manipulation approaches, such as optical tweezers using tightly focused laser beams or strong evanescent fields at total-reflection interfaces[11,12], can only be effectively implemented by eliminating the surface adhesion in liquid environments. Nevertheless, since the final working environments of most integrated photonic circuits are in air or vacuum[13–15], the influence of fluid convection, disturbance, and surface tension will severely limit the integration accuracy during the removal of liquid from the assemblies[16]. Hence, there is an increasing need for a method that enables on-chip integration directly in non-liquid environments.

Precisely manipulating single metal nanowires along the waveguides is of essential importance in on-chip integration, such as adjusting the relative coupling positions and lengths between the metal nanowires and the dielectric waveguides[7,17,18]. However, to the best of our knowledge, no such manipulation of metal nanowires in non-liquid environments has been reported so far. Currently, a common manipulation method in air environments is manually using 3D stage-actuated tungsten or silica probes to apply thrust on the nanowire sides[6,7,18,19], but the nanowires can only be moved laterally with accuracy of ~1 μm. On the other hand, surface acoustic waves (SAWs) induced by the elastic expansion of metal lattices under transient heating of pulsed lasers[20–22], have been used to drive microscale metal objects for surface cleaning, particle detaching, and microplate rotating[23–25], thus providing the possibility of overcoming the strong surface adhesion to manipulate metal nanowires in non-liquid environments.

Here, we propose an earthworm-like peristaltic crawling motion mechanism based on the synergistically working of expansion, friction, and contraction, and experimentally demonstrate continuous and controllable manipulation of single metal nanowires on fixed microfibres in non-liquid environments by plasmon driving, with advantages of sub-nanometer positioning accuracy, low actuation power, and self-parallel parking. Capitalizing on this approach, we further perform on-chip manipulations of single nanowires on fixed microfibres in hybrid photonic-plasmonic circuits including transporting, positioning, coupling, and sorting, to demonstrate the advantages of on-situ operation, high selectivity, and great versatility.

## Results

**Plasmon-driven nanowire actuators.** As illustrated in Fig. 1a, two gold (Au) nanowires are placed on two suspended silica

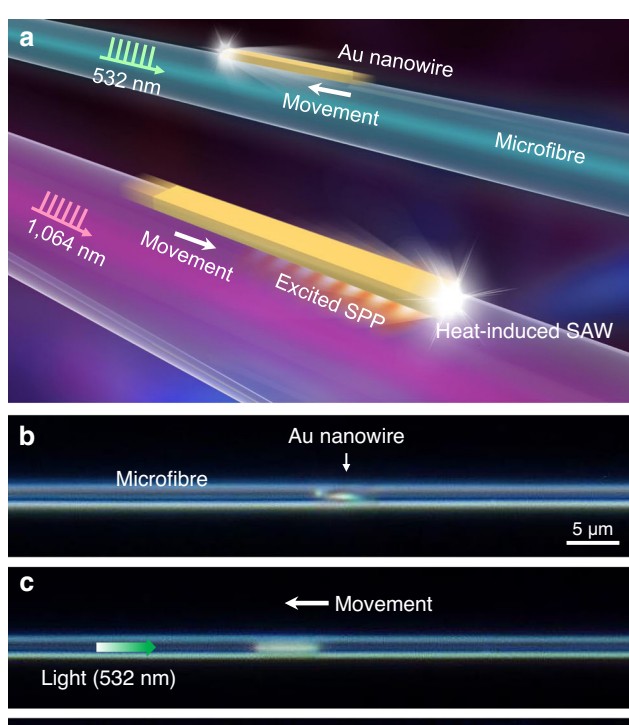

**Fig. 1 Plasmon-driven movements of Au nanowires on silica microfibres. a** Schematic showing movements of Au nanowires on suspended microfibres, driven by pulsed lasers with wavelengths of 532 and 1064 nm, respectively. Only the 1064 nm laser can efficiently excite SPP in the Au nanowire, thus enhancing the heating effect of absorbed light and the induced SAW drive the Au nanowire to peristaltically crawl along the silica microfibre. **b** Microscope image of an Au nanowire ($L_{NW} = 6.1$ μm) placed on a suspended microfibre ($D_{fiber} = 1.8$ μm). Note that the nanowire axis was not strictly parallel with the microfibre axis. **c** Under the 532 nm laser actuation from the left side, the Au nanowire moved to the left. **d** Under the 1064 nm laser actuation from the left side, the Au nanowire moved to the right. The used average power of the 532 and 1064 nm pulsed lasers was about 40 μW.

microfibres, and two pulsed lasers with different wavelengths are launched into the microfibres as the actuation sources. Experimentally, Fig. 1b shows an Au nanowire with a length ($L_{NW}$) of 6.1 μm placed on a suspended and fixed microfibre with a diameter ($D_{fiber}$) of 1.8 μm (see "Methods" section and Supplementary Fig. 1). The surface adhesion can tightly bind the nanowire on the microfibre surface and was experimentally measured to be about 2 μN (Supplementary Note 1). When a 532 nm nanosecond pulsed laser was launched into the microfibre from the left side with average power of 40 μW (corresponding to single-pulse energy of 10 nJ), the Au nanowire moved to the left (Fig. 1c); on the contrary, when a 1064 nm nanosecond pulsed laser was launched from the left side also with average power of 40 μW, the Au nanowire moved to the right (Fig. 1d). This phenomenon, i.e., the movement directions of the nanowires and the light propagation directions are opposite for the 532 nm light, while being the same for the 1064 nm light, were observed in dozens of Au nanowire samples, which had lengths ranging from 1 to 25 μm and were placed on different microfibres with diameters ranging from hundreds of nanometers to 3 μm (Supplementary Fig. 2). In addition, as increasing the repetition rates of

the pulsed lasers, the moving speeds of the nanowires also increased linearly. For nanowires with much larger cross-sections (Supplementary Fig. 3) or longer lengths (Supplementary Fig. 4), their moving speeds decreased compared with those with shorter lengths or smaller cross-sections, which can be attributed to the increased adhesion force to the microfibre surface. But for much shorter nanowires with lengths less than ~1.5 μm, no detectable movement was observed (Supplementary Fig. 4).

Such light-driven phenomena were also observed in metal nanowires of other materials such as palladium nanowires[7,26] (Supplementary Fig. 5). Under special environmental conditions such as high vacuum (less than $10^{-4}$ Pa) and low temperature (down to 30 K), such phenomena were still observed, indicating that the photophoretic force[25,27,28] induced by the surrounding gas molecules can be excluded. In addition, no recrystallization phenomena in as-driven Au nanowires were observed by using a high-resolution atomic force microscope (AFM) and a transmission electron microscopy (Supplementary Fig. 6). Other laser sources (see Methods) were also used to investigate the nanowire movement behaviors. Similar movements were also observed by using a supercontinuum source with pulse duration of ~1 ns (Supplementary Movie 1). However, no detectable nanowire movement was observed when continuous-wave lasers were used, even though the laser power was high enough to damage the nanowires, indicating that the optical force induced by the photon momentum can also be excluded[28]. Although 1,035-nm-wavelength picosecond (pulse duration: ~6 ps) and femtosecond pulses (pulse duration: ~600 fs) could drive the nanowires to move a few micrometers, the nanowires would quickly stop moving. Then even if the laser power was increased until the nanowires were damaged, the nanowires still could not continue to move. AFM scans reveal that the surface roughness of as-driven nanowires increased a lot (Supplementary Fig. 7), suggesting a recrystallization or reconstruction effect on the surface.

When a metal nanowire is placed on a microfibre surface, the high-fractional evanescent fields outside the microfibre will strongly interact with the metal nanowire. By using a three-dimensional finite-difference time-domain (3D-FDTD) method (Supplementary Note 2), Fig. 2a, b show the electric field intensity distributions of a nanowire–microfibre hybrid structure, which are excited by the $HE_{11}^{x}$ and $HE_{11}^{y}$ fundamental modes of guided 1064 nm light in the microfibre, respectively. The Au nanowire has a thickness of 100 nm, a width of 200 nm, and a length of 6.0 μm, and the microfibre has a $D_{fiber}$ of 1.8 μm. The highly concentrated intensity and apparent standing-wave patterns along the nanowire–microfibre interface observed in Fig. 2a, confirm that SPPs can only be well excited by the $HE_{11}^{y}$ mode at the 1064 nm wavelength. But for both the $HE_{11}^{y}$ and $HE_{11}^{x}$ modes of guided 532 nm light, no obvious SPP excitation can be observed around the nanowire–microfibre interface (Supplementary Fig. 8). Therefore, although both the 532 and 1064 nm lasers can drive the nanowires, only the driven at the 1064 nm wavelength can be called plasmon driven, as noticed in Fig. 1.

The absorptance spectra of the Au nanowire shown in Fig. 2c further confirms that the $HE_{11}^{y}$ modes in longer wavelength ranges can induce much stronger absorptance due to the efficient SPP excitation (Supplementary Fig. 9 and Supplementary Note 2). In addition, compared to the guided light field confined inside the microfibre with $D_{fiber} = 1.8$ μm, the 1064 nm excited SPP field around the nanowire–microfibre interface in Fig. 2b has an effective mode area of about 0.001 μm$^2$, thus enhancing the electric field intensity more than six times and dramatically increasing the light absorption by the nanowire. The absorbed light by Au nanowires is subsequently turned into thermal energy. For the $HE_{11}^{x}$ modes (Fig. 2d, f), the heat power density along the Au nanowire bottom (parallel to the microfibre axis) is

approximately evenly distributed along the nanowire bottom. Nevertheless, this even distribution without significant gradient changes result in evenly distributed friction throughout the bottom (as discussed below), so that the nanowire cannot be driven along the microfibre. For the $HE_{11}^{y}$ modes guided in the microfibre (Fig. 2e, f), due to the relative low propagation loss of SPPs at 1064 nm, the guided light propagates forward synchronously with the excited SPPs, and simultaneously excites more new SPPs, resulting in a continuous increase in the SPP energy along the nanowire bottom (Supplementary Fig. 10 and Supplementary Movie 2). When reaching the nanowire frontend (with respect to the guided light direction), the accumulated forward propagating SPPs will be reflected by the nanowire frontend and propagate backward, thus forming a standing wave at the nanowire–microfibre interface. Due to the existence of propagating loss, the intensity of the SPP standing wave reaches the maximum at the nanowire frontend and then decays toward the backend. The SPP standing wave will further enhance the heating effect at the nanowire frontend, and such a gradient distribution of the heat power density is beneficial for the nanowire movement as discussed below. Moreover, for shorter nanowires with lengths less than ~1.5 μm, the SPP excitation efficiency of the nanowire by the microfibre will decrease and the generated heat power density is much smaller than those in longer nanowires (Supplementary Fig. 4), causing shorter nanowires unable to overcome the adhesion force and move.

**Peristaltic crawling motion mechanism of nanowire actuators**. The generated thermal energy can cause the lattice of Au nanowires to expand, and when the resulting SAW energy is sufficiently strong, Au nanowires can overcome the surface adhesion and move forward. Here we propose an earthworm-like peristaltic crawling motion mechanism based on the synergistically working of expansion, friction, and contraction[29]. As illustrated in Fig. 3a, firstly, the SPP excited by a 1064 nm pulsed laser causes the nanowire frontend to expand, and then the friction between the nanowire and the microfibre increases due to the reduction of the interfacial gap (Supplementary Note 1). Finally, as the nanowire contraction begins, the nanowire center is pulled forward. The calculation in Fig. 3b, c shows that in the $y$ direction, the initial transient thermal excitation causes the upper and lower surfaces of the nanowire frontend to vibrate in a symmetric plate wave mode (Supplementary Note 2 and Supplementary Movie 3). In particular, the lower surface reaches a maximum downward expansion of 0.4 nm in the first 6 ns. The reduced interfacial gap (associated with surface roughness and van der Waals forces, typically at a scale of ~1 nm[10,30], also see Supplementary Fig. 1), thereby makes the frontend the most adhesive of the entire nanowire bottom. Meanwhile, in the $z$ direction (Fig. 3d, e), the nanowire frontend reaches a maximum forward elongation of 6.4 nm around the 11th ns and subsequently begins to contract. Therefore, with the nanowire frontend tightly adhered to the microfibre surface, the lattice contraction can overcome the surface adhesion of the rest of the bottom part and pull the entire nanowire forward.

This forward crawling behavior will continue for more than 30 ns, during which the nanowire moves forward by a distance of 1.07 nm (Fig. 3e). After 40th ns, the upper and lower surfaces of the nanowire frontend begin to vibrate in an antisymmetric plate wave mode with greatly reduced amplitude (Fig. 3c), and also the entire nanowire is in a state of balanced vibration. Under this condition the friction is evenly distributed throughout the bottom, and thus the contraction no longer causes the nanowire center to move. By the repetition of this cycle, the nanowire crawls forward in a step-by-step manner. At the same time of

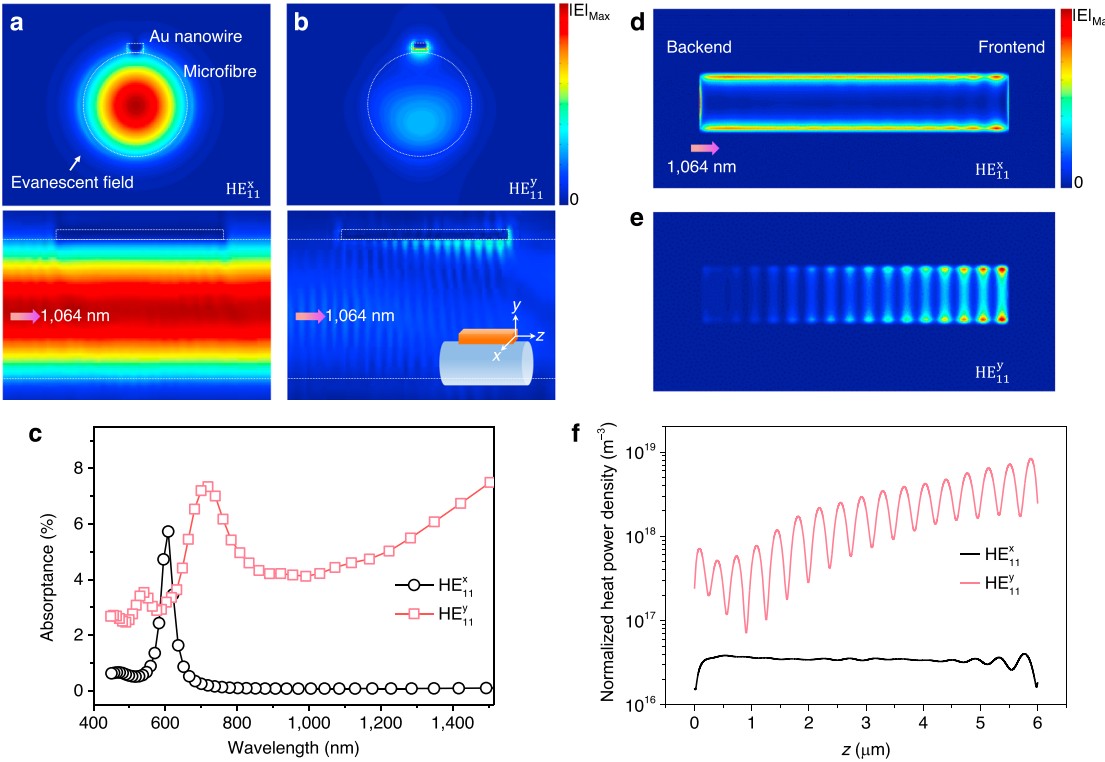

**Fig. 2 Calculated distributions of electric fields and heat power densities in Au nanowires. a, b** Electric field distributions of the nanowire–microfibre system at the (upper images) transverse and (lower images) longitudinal sections, excited by **a** $HE_{11}^{x}$ fundamental mode and **b** $HE_{11}^{y}$ fundamental mode guided in the microfibre at 1064 nm wavelength. The Au nanowire has a thickness of 100 nm, a width of 200 nm, and a length of 6.0 μm, and the microfibre has a $D_{fiber}$ of 1.8 μm. Inset of (**b**) lower image shows the definition of electric field directions. **c** Absorptance spectra of the Au nanowire with different fundamental modes. **d, e** Spatial distributions of heat power density on the Au nanowire bottom surface for **d** the $HE_{11}^{x}$ modes and **e** the $HE_{11}^{y}$ modes, respectively. **f** Heat power density value along the sideline (z direction) of the nanowire bottom surface.

the SAW propagation, most of the thermal energy at the nanowire frontend is transferred to the entire nanowire through thermal conduction until the heat distribution along the entire nanowire is balanced. The calculated peak of the transient temperature response at the nanowire backend is about 890 K (Fig. 3f and Supplementary Note 2), and the maximum thermal expansion of the nanowire in the z direction is estimated to be 0.94 nm, which agrees with the results in Fig. 3d, e. By fitting the attenuation part of the response curve, a time constant of about 12 ns is obtained. Due to the long length of nanowires and the propagation of surface acoustic waves, the equilibrium time required for thermal conduction and mechanical oscillation is much longer than that of heat dissipation in nanoparticles and nanorods[31–33].

From the motion mechanism proposed above, the process of heat induced lattice expansion is an essential initial step for the nanowire movement, and in principle, as long as the lattice expansion can occur, our driving mechanism can be applied to other light sources to drive the nanowires. However, for picosecond (less than ~200 ps) and femtosecond pulses, their high peak energy can induce obvious unbalanced spatial-temporal temperature response between the center and side points of the nanowire frontend at the beginning of the light absorption (Supplementary Note 3). In such an ultrashort time scale, the heat transfer from the side point to the center point could not been completed, and the highly concentrated heat at the side point may damage the nanowire lattice. So to obtain continuous and controllable manipulation, the pulsed lasers with duration around the nanosecond scale are the very effective sources to drive the metal nanowires.

In addition, the heating positions formed at the frontend or backend of the nanowire will determine the movement direction.

For shorter wavelengths such as the 532 nm wavelength (Fig. 1c), due to the strong absorption of Au material to light, no obvious SPP is excited and propagates in the nanowire, thus no standing wave will be formed around the nanowire bottom (Supplementary Movie 2). Only the direct thermal absorption of the $HE_{11}^{y}$ mode light (Supplementary Fig. 11) generates a heating source around the nanowire backend, and thus the nanowire will be driven to the opposite direction of the light. It is also noticed that the maximum magnitude of the heat power density induced by the 532 nm light is an order of magnitude lower than that induced by the 1064 nm light, which is attributed to the absence of the SPP standing wave. So experimentally, much higher power of the 532 nm nanosecond laser is usually needed to drive the nanowires, which may bring damage to the nanowires. Therefore, we used the 1064 nm pulsed laser for practical manipulating metal nanowires.

**Controlling nanowire manipulation.** It is noticed that in Fig. 1b, the initial position of the Au nanowire axis was not strictly parallel with the microfibre axis, but after moving along the microfibre, the nanowire axis gradually turned parallel to the microfibre axis. To make this phenomenon more clearly, Fig. 4a shows a main process that when driven by the 1064 nm laser with average power of 6 μW, an Au nanowire ($L_{NW} = 6.3$ μm) tilted on a suspended microfibre ($D_{fiber} = 2.1$ μm) gradually rotated clockwise until parallel to the microfibre axis, during which the nanowire simultaneously moved forward (Supplementary Movie 4). As shown in Fig. 4b, for the initial posture with a tilted angle of 22°, the heating sources are mainly distributed at the middle and right portions of the nanowire, and the heat power density at the lower side is stronger than that at the upper side. Such uneven

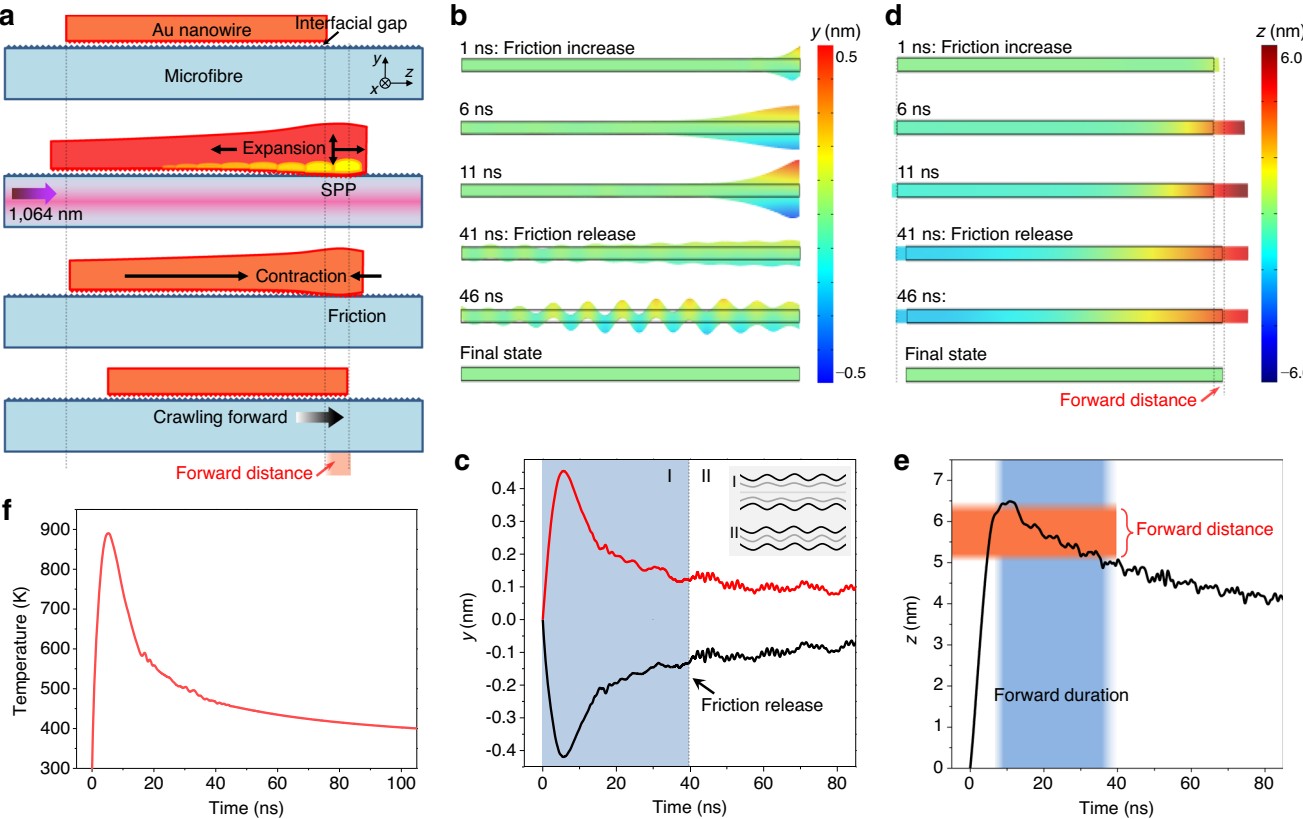

**Fig. 3 Peristaltic crawling mechanism of the Au nanowire actuator. a** Schematics of the peristaltic crawling of an Au nanowire on a silica microfibre. The SPP excited by a 1064 nm pulsed laser causes the nanowire frontend to expand and increases its friction, thus pulling the entire nanowire forward as the nanowire contraction begins. The forward distance is denoted with an orange background. The dimensions of the nanowire and the microfibre are the same as those in Fig. 2. Note that there is an interfacial gap at a scale of ~1 nm between the nanowire and microfibre. **b, d** Time-course changes of the nanowire in the $y$ (**b**) and $z$ (**d**) directions upon laser actuation. **c** Time-dependent vibration behaviors of the upper and lower surfaces of the nanowire frontend, which show a mode of a symmetric plate wave before 40th ns (denoted with a blue background). Inset illustrates the (I) anti-symmetric and (II) symmetric plate wave. **e** Time-dependent axial expansion behavior of the nanowire frontend. The ranges of forward distance and duration are denoted with orange and blue backgrounds, respectively. **f** Calculated time-dependent transient temperature change at the nanowire frontend.

distributions will cause the nanowire to rotate clockwise until the nanowire axis is parallel with the microfibre axis. This rotation phenomenon, which can be called self-parallel parking, can be used to adjust the initial postures of metal nanowires on microfibres, which facilitates on-chip manipulation processes.

The movements of Au nanowires are then precisely controlled by tuning the repetition rates of the 1064 nm pulsed laser. By maintaining each pulse energy constant at 8.6 nJ, Fig. 4c shows a series of photographs of an Au nanowire ($L_{NW} = 2.3$ μm) moving on a suspended microfibre ($D_{fiber} = 2.2$ μm) as the laser repetition rates decreasing from 1600 Hz to 50 Hz. Figure 4d summarizes the dependence of the moving speed of the nanowire on the laser repetition rates. According to the slope of the fitted trend line, the calculated positioning resolution of single-pulse actuation is 0.56 nm, which is comparable to the resolutions of commercial ultrafine piezoelectric actuators. The calculated peak of the transient temperature response at the nanowire backend is about 400 K (Supplementary Fig. 12), and the maximum expansion can be estimated as 0.17 nm, which agrees with the experimental result. Besides, the calculated moving speed is 6.5 μm s$^{-1}$ per milliwatt of the 1064 nm laser power, which is two orders of magnitude larger than the micro/nanoparticle transport speed using optical tweezers and near-field evanescent forces (actuation average power typically exceeds tens of milliwatts)[24,34,35], suggesting the higher efficiency of our plasmon driven approach.

**On-chip manipulation in integrated photonic circuits.** Manipulating single metal nanowires in non-liquid environments with high-precision controllability open the door to practical on-chip integration. For example, Fig. 5a shows a schematic prototype of a photonic chip that contains typical components such as couplers, resonators, and interferometers[13–15]. We first demonstrate some basic on-chip manipulation of the Au nanowires on the fixed microfibres such as transport, positioning, and orientation. Curved waveguides such as the U-shaped or C-shaped structures are essential components for constructing couplers and resonators. Figure 5b shows the transport of an Au nanowire ($L_{NW} = 3.8$ μm) along a C-shaped curved microfibre ($D_{fiber} = 2.3$ μm) until it reached the distal end of the microfibre (Supplementary Movie 5). Controlling the propagation length of SPPs[4–7] is very important to tune the property of metal nanowire-based plasmonic devices. Figure 5c shows the suspended part of an Au nanowire ($L_{NW} = 3.8$ μm) placed on the tip of a fiber taper gradually increased by plasmon driving. When a 660 nm signal laser was launched into the microfibre, we can see that the brightness of emission spots at the nanowire end decreases as the suspended nanowire length increases[17,18]. In addition, different moving orientations of individual metal nanowires can be achieved by selectively actuation. As shown in Fig. 5d, three Au nanowires were placed on a fixed microfibre ($D_{fiber} = 2.1$ μm) and identified by numbers of 1, 2, and 3, respectively. We first used a fiber taper to

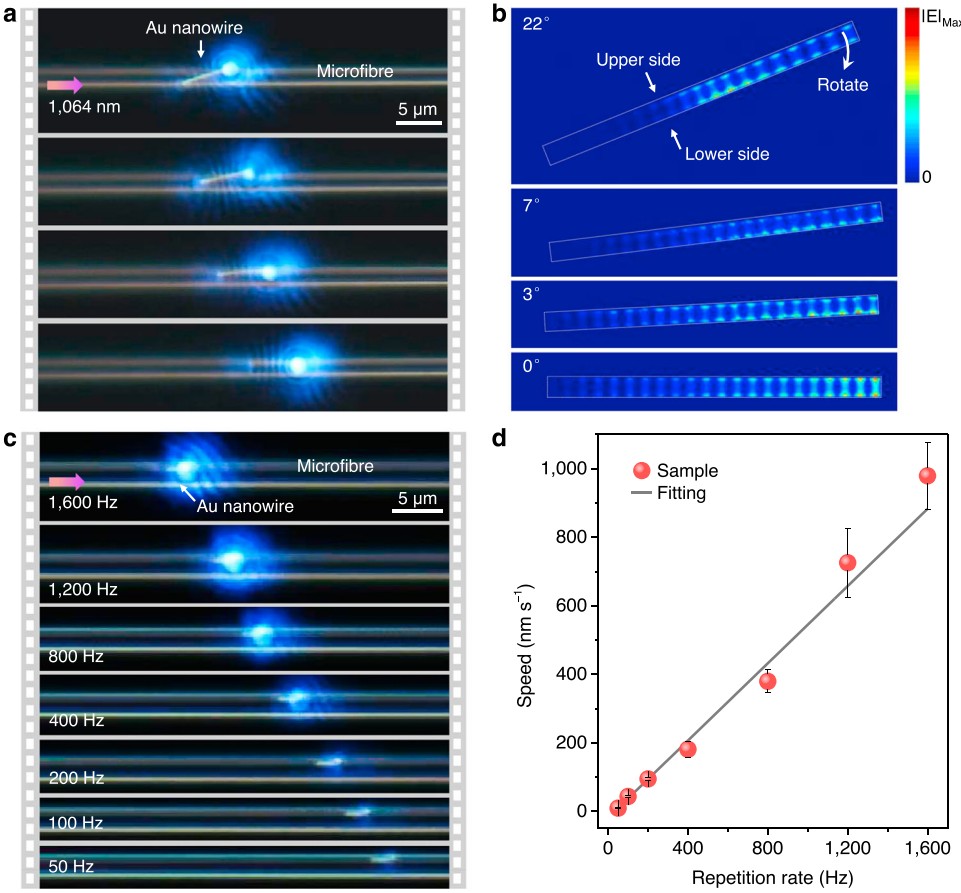

**Fig. 4 Controlling manipulations of Au nanowires on microfibres. a** Sequencing photographs of an Au nanowire ($L_{NW} = 6.3$ μm) tilted on a suspended microfibre ($D_{fiber} = 2.1$ μm) gradually rotating to the parallel position, driven by the 1064 nm laser with average power of 6 μW. **b** Simulated heat power density distribution on the nanowire bottom with different tilted angles of 22, 7, 3, and 0°. **c** Sequencing photographs of an Au nanowire ($L_{NW} = 2.3$ μm) moving on a suspended microfibre ($D_{fiber} = 2.2$ μm) as decreasing the laser repetition rates from 1600 to 50 Hz, during which each pulse energy remained constant at 8.6 nJ. **d** Dependence of the moving speed of the nanowire on the laser repetition rates. Error bars are the variance of the moving speeds.

couple the 1064 nm laser into the microfibre from the left side of the nanowire 2, and actuated the nanowire 2 to move toward the right by a distance of ~2 μm. Then we coupled the 1064 nm laser from the right side of the nanowire 1 and actuated the nanowire 1 to move toward the left by a distance of ~5 μm. During both processes, the nanowire 3 remained stationary.

Next we demonstrate the manipulation of single nanowires between multiple fixed microfibres. In a coupler composed of two side-coupled microfibres shown in Fig. 5e, an Au nanowire ($L_{NW} = 6.6$ μm) was transported from the left microfibre ($D_{fiber} = 2.4$ μm) to the right one ($D_{fiber} = 1.8$ μm). Similar to the sorting of the micro/nanoparticles using a Y-branch waveguide in the liquid by evanescent force[12,36], we can also selectively control the movement of the metal nanowires on different microfibres. As shown in Fig. 5f, when the laser was launched into the upper microfibre ($D_{fiber} = 2.0$ μm), an Au nanowire moved to the coupling area of two microfibres until it contacted the lower microfibre ($D_{fiber} = 2.1$ μm). When the laser was launched into the lower microfibre, the Au nanowire was attracted onto the lower microfibre, and then we can control the nanowire movement directions along the lower microfibre by coupling the actuation laser from the left or the right.

## Discussion

In summary, we have demonstrated a facile plasmon-driven approach to continuously and controllably manipulating single

metal nanowires along fixed silica microfibres in non-liquid environments. The SPPs in Au nanowires are effectively excited by the high-fractional evanescent fields outside the silica microfibres, and form standing waves around the nanowire frontends, which enhances the heating effect of absorbed light and is beneficial for the nanowire movement. The heating-induced SAWs overcome the strong surface adhesion and actuate the nanowires to peristaltically crawl forward, with a positioning accuracy as small as 0.56 nm. In addition, our plasmon-driven approach can also adjust the initial postures of metal nanowires on microfibres and will facilitate the on-chip manipulation processes. In principle, our demonstrated manipulation mechanism can also be applied to fixed photonic integrated circuits constructed with other materials and structures such as silicon and silicon nitride waveguides[13,14,37]. In past years, metal nanowires have been integrated with various micro/nanowaveguides to realize various functionalized circuits and devices ranging from routers/couplers, interferometers, and resonators, to lasers[15]. Looking forward, our plasmon-driven approach could be combined with other nanowire manipulating approaches[11,12,19] to work synergistically, and in this way, we might realize co-integration of various functionalized photonic components on single chips.

## Methods

**Fabrication of silica microfibres and metal nanowires.** Silica microfibres and fiber tapers used in this work were drawn from standard fibers (SMF-28, Corning) by using a simple flame-heated method[38,39]. The diameters of microfibres were in

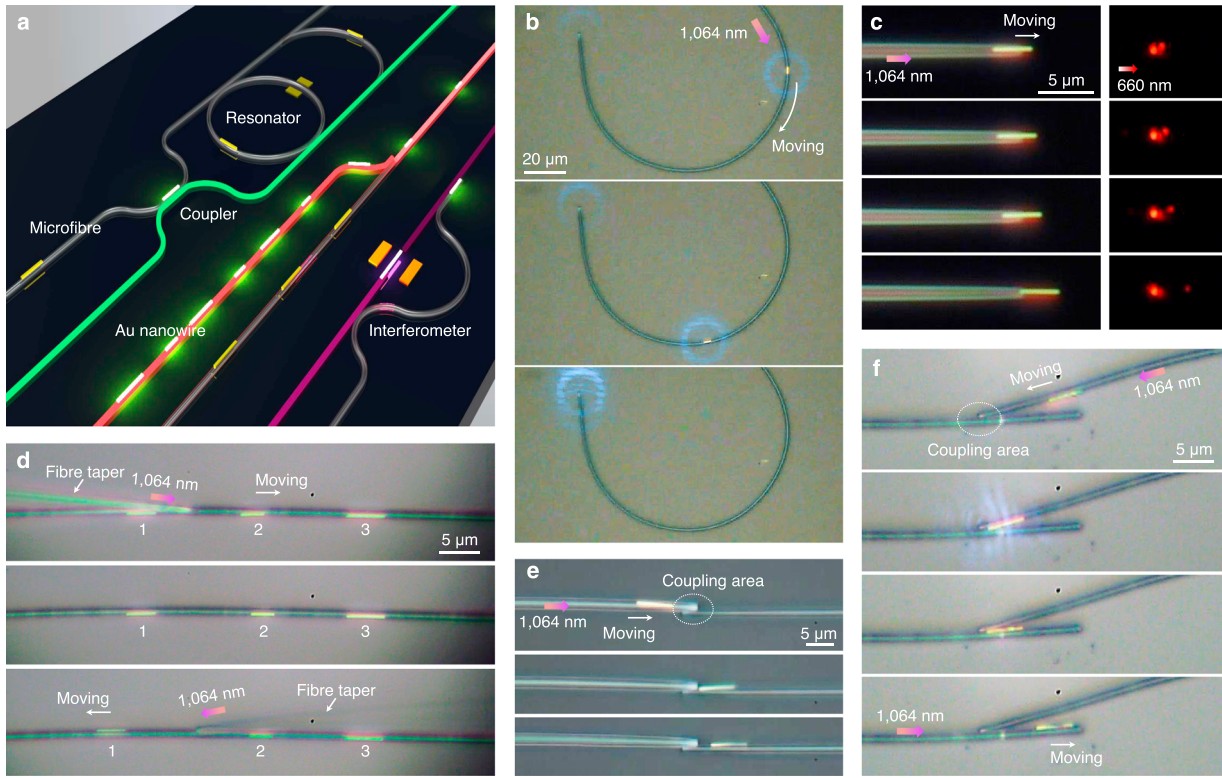

**Fig. 5 On-chip manipulation of Au nanowires in integrated photonic circuits. a** Schematic prototype of a typical photonic chip that contains typical components such as coupler, resonator, and interferometer. **b** Transport of an Au nanowire ($L_{NW} = 3.8$ μm) along a C-shaped curved microfibre ($D_{fiber} = 2.3$ μm). **c** Controlling the propagation length of SPPs in an Au nanowire ($L_{NW} = 3.8$ μm) placed on the tip of a fiber taper. A laser with a wavelength of 660 nm was launched into the microfibre as the signal light. **d** Selectively actuation of individual Au nanowires on a microfibre ($D_{fiber} = 2.1$ μm) by using a fiber taper to couple the actuation laser. **e** Transport of an Au nanowire ($L_{NW} = 6.6$ μm) from the left microfibre ($D_{fiber} = 2.4$ μm) to the right microfibre ($D_{fiber} = 1.8$ μm). **f** Sorting of an Au nanowire ($L_{NW} = 3.8$ μm) using two contact microfibres. All actuation lasers used here are the 1064 nm pulsed lasers with average input power ranging from ~20 to 300 μW.

the range of 1–3 μm. Single-crystal Au nanowires used in our work were fabricated by a simple vapor transport method[7,26]. Vertical Au nanowires were grown on a *c*-cut sapphire substrate at a source temperature of 1200 °C. High-purity argon gas was used as the carrying gas and the growth time is about 40 min. As-fabricated Au nanowires had rectangular cross-sections and smooth surfaces (Supplementary Fig. 1), and the Au nanowires used in this work had widths and thickness around from 100 to 300 nm. Silica microfibres were first placed on or protruded from the edge of a MgF$_2$ substrate. For robust operation, microfibres were fixed on the MgF$_2$ substrate using a low-index UV cured fluoropolymer (EFIRON PC-373; Luvantix Co. Ltd.). As-fabricated Au nanowires were picked up from the grown substrate via mechanical micromanipulation and placed on the microfibres (Supplementary Fig. 13).

**SPP excitation and nanowire actuation**. Experimentally, we used several laser sources to investigate the nanowire movement behaviors. The used 1064 nm wavelength nanosecond pulsed laser was from a solid state Q-switched laser (Changchun New Industries Optoelectronics Tech. Co., Ltd.), with pulse duration of 8.5 ns and tunable repetition rates from 1 Hz to 4 kHz. The used 532 nm pulsed laser was generated from the 1064 nm nanosecond laser by using a frequency-doubling KTP crystal, with pulse width of 5.6 ns and also tunable repetition rates. The pulsed lasers were linearly polarized with TEM$_{00}$ transverse mode. The used supercontinuum source (SuperK Compact, NKT photonics) with master source pulse width of ~1 ns and a constant repetition rate of 24 kHz, was filtered by a 664 nm long-pass filter to obtain light in the longer wavelength range. The used 1035 nm-wavelength pulsed fiber laser has tunable pulse duration from ~400 fs to 6 ps and a repetition rate of 25 kHz (FemtoYL-50, Wuhan Yangtze Soton Laser Co., Ltd.) A 660 nm continuous-wave laser was used as a signal light in Fig. 5c. The excitation light was first lens-coupled into a standard silica fiber (SMF-28, Corning) and then squeezed into the tapered microfibre to excite the SPPs of metal nanowires (Supplementary Fig. 14). Because tapered microfibres do not support guiding modes with linear polar-ization, the linearly polarized light coupled into the microfibre will finally evoke a quasi-circular or elliptical polarization of the guided modes (HE$_{11}$ modes). The movements of the metal nanowires on microfibres were monitored using a 100× microscope objective and a CCD camera. To investigate the nanowire

movement in special environmental conditions such as high vacuum (less than $10^{-4}$ Pa) and low temperature (down to 30 K), the Au nanowires and micro-fibres were placed in a cryostation from Montana Instruments (3–350 K).

**Data availability**
The data that support the findings within this study are available from the corresponding author upon reasonable request.

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

## Acknowledgements

This work is supported by the National Natural Science Foundation of China (11674230) and the Shanghai Rising-Star Program (18QA1403200).

## Author contributions

F.G. conceived the idea and designed the research. S.L. performed the experiments, collected and analysed the data, and wrote the paper; Z.G. performed the theoretical calculations and analysis, and wrote the paper; R.Z. and J.P. contributed in material fabrication and data analysis; J.L., W.F., Z.Y., H.Y., Z.L., Q.Z., S.Z., and M.G. helped in data analysis and manuscript writing. All authors discussed the results and commented on the manuscript.

## Competing interests

The authors declare no competing interests.
