## [Peer Review File · Nature Communications]

REVIEWER COMMENTS

Reviewer #1 (Remarks to the Author):

The authors reported an exciting new method that enables the precise manipulation of chemically synthesized plasmonic nanowires in a non-liquid environment and their on-chip integration with photonic components. Utilizing the periodic local plasmonic heating, friction pinning and cooling contraction, they were able to achieve a series of precise manipulations of a single gold nanowire. The authors propose a convincing mechanism for the plasmon-driven movement and provide detailed analysis to the response to different wavelength and frequency of the excitation. Although the method is still on a single wire basis and not yet fit for large scale fabrications, it still marks an important step towards the rational integration of photonic circuit components from top-down lithography and high quality plasmonic nanowires from bottom-up chemical synthesis, which has been a long-standing obstacle for the application of plasmonic nanowires in photonic circuitry. I would happily endorse it's publication in Nature Communications when the following issue is addressed:

The major motivation of incorporating chemically synthesized plasmonic nanowires instead of their microfabricated counterparts into the photonic circuit, is their high crystallinity and the resulting low propagation loss. So it is important that the positioning method would not negatively impacts the crystallinity and plasmonic loss. The manuscript estimates transient temperature response at the nanowire backend peaking about 890 K, which would likely result in recrystallization. The authors should address how much the repeated heating-cooling cycle affect these intrinsic characteristics of the chemically synthesized gold nanowire. This can be done by measuring the propagation length of the nanowire before and after moving a certain length. Electron diffraction pattern on different part of the wire would also helpful to understand if the single crystallinity is preserved.

Reviewer #2 (Remarks to the Author):

In the paper entitled 'Plasmon-driven nanowire actuators for on-chip manipulation' the authors manipulate plasmonic gold nanowires with dielectric waveguides by applying laser pulse in the waveguides. When applying 532 nm light, the wire will be withdraw to the light source direction and when applying 1064 nm light, the wire will be driven to the light propagating direction because the plasmon induced surface acoustic wave make the wire crawl like a earthworm . Even if the wires have crossing angle with the fiber, the optical force will make the wire self-parallel parking through the fiber. With this technique, the authors demonstrated that they can manipulate the gold wires through U, C shaped fibers and transfer a wire to another fiber. This technique has potential use on the plasmonic wire based circuit. The technique is quite advanced and practical and the fiber induced surface acoustic wave here is very interesting. I think this work will attract the interest of broad readers and will bring impact in the future. So I recommend it for publication in nat. comm. However, there are some critial points need to be addressed (the importance is randomly listed).

1. In the paper, the authors said that there was no detectable nanowire movement observed when continuous-wave lasers or femtosecond pulsed lasers were used, even though the laser power was high enough to damage the nanowires, indicating that the optical force induced by the photon momentum can also be excluded. So I conclude that the pulse width, repeating frequency are very important parameters for the wire driven technique. However, the authors didn't give any focus on the pulse width. it is only mentioned in the method part for 1064 nm light. For the doubled frequency 532 nm, if the with changed a lot? The authors should give proper thoeretical or experimental prove together with the discussion in the main text because the acoustic wave of plasmonic wire should be considered (the decay time of a 100 nm particle is about 100 ps). For the repetition frequency, it is easier to understand that the wire moving speed should be linearly

related to it. But the repetition should come earlier in the main text.

2. For the movement detail, it is not explained quite clear in the paper. There have been many papers on the plasmon decayed lattice oscillation researching. For example, for a about 100 nm particle, usually the plasmons will decay in about 10s fs and transfer the energy to the lattices in about 1 ps time. Then there is acoustic wave oscillation and in about 100 ps, the oscillation will arrive at the balance. In this work, consider the length of the wire, the plasmons have much faster speed than the acoustic speed on the wire, and the lattice oscillation should happen almost the same time in the whole wire. But from the supporting information I see that the waves (from the main text I think it is the lattice oscillation waves) propagate to the end in about tens of nano seconds and the oscillation keeps over 100 ns. So what really happens here is not clearly addressed in the paper.

3. Continue with the above comment, what happens if changing the pulse duration time? Maybe the authors don't have the instrument to sweep this parameter, but the authors have already done it with fs lasers, so I think at least for ps laser it should be included here for the mechanism.

4. Consider above comment, in 1 ns pulse duration, the wire should be already thermalized balantly. If not, what is really the mechanism here.

5. Consider above comment, for difference length of wires, the movement speed should be different because the heating pulse takes more time to get the balance. The authors have done experiments for different length of wires, but didn't give any statistics or comment on it.

6. Continue with the above comment, for the simulations, in FDTD method only pulse is used to excite samples and different wavelength post results are got from fourier transformation. So in 1 ns duration, the light has already oscillated for about 10^5 times, the heat effect should be an averaged value. For finite wire, there is standing wave, so the heating can be wave like distribution. Considering the energy dissipating, the heating energy in one end should be large than the other end. But if the SPPs dissipate along the wire very fast, there will be not standing wave. How did the authors get the thermal waves in the paper (especially Figure 2e)? I didn't find it in the methods part.

7. For the simulations, the authors didn't give the pulse duration information. If 1 ns is used here, then the covered wavelength range should be very limited. If it is available for the simulations.

8. In Figure S3, the authors gives the E field distributions of the wire-fiber. From the figure one can see that the wire has little influence for the fiber. So why is the mode distribution an elliptic area but not a circular area? and From Figure S3b right column I see that the boundary is almost flat in the whole fiber no matter if there is wire or not.

9. In figure 2, why the E field on the right side of the wire is stronger should be explained in the text.

10. Is there any gradient force here in Figure 2?

11. In the paper the authors didn't give the mechanism of the pulling back effect for 532nm. Even if the title of the paper is 'plasmon driven', the pulling back effect for 532 nm should be explained in the paper. Otherwise the work is not complete.

12. Indeed this method can manipulate the wire with very accurate steps along the wire, but the position accuracy also depends on the fiber position. And the fiber position is still controlled through the scanning stage or other mechanical methods. So in the introduction and conclusion part, the authors said for accurately control the wire for optical chip is a little bit overselling and should be modified.

13. In Figure S4a, why is the heating distribution is so asymmetric?

14. The effect of the diameter of the wire should be discussed.

Response Letter to *Nature Communications* (Manuscript No.: NCOMMS-20-12006)

Title: "Plasmon-driven nanowire actuators for on-chip manipulation"

Author(s): Shuangyi Linghu, Zhaoqi Gu, Jinsheng Lu, Wei Fang, Zongyin Yang, Huakang Yu, Zhiyuan Li, Runlin Zhu, Jian Peng, Qiwen Zhan, Songlin Zhuang, Min Gu & Fuxing Gu

Response to Reviewer's comments:

Reviewer #1

The authors reported an exciting new method that enables the precise manipulation of chemically synthesized plasmonic nanowires in a non-liquid environment and their on-chip integration with photonic components. Utilizing the periodic local plasmonic heating, friction pinning and cooling contraction, they were able to achieve a series of precise manipulations of a single gold nanowire. The authors propose a convincing mechanism for the plasmon-driven movement and provide detailed analysis to the response to different wavelength and frequency of the excitation. Although the method is still on a single wire basis and not yet fit for large scale fabrications, it still marks an important step towards the rational integration of photonic circuit components from top-down lithography and high quality plasmonic nanowires from bottom-up chemical synthesis, which has been a long-standing obstacle for the application of plasmonic nanowires in photonic circuitry. I would happily endorse its publication in *Nature Communications* when the following issue is addressed:

Comment: *The major motivation of incorporating chemically synthesized plasmonic nanowires instead of their microfabricated counterparts into the photonic circuit, is their high crystallinity and the resulting low propagation loss. So it is important that the positioning method would not negatively impacts the crystallinity and plasmonic loss. The manuscript estimates transient temperature response at the nanowire backend peaking about 890 K, which would likely result in recrystallization. The authors should address how much the repeated heating-cooling cycle affect these intrinsic characteristics of the chemically synthesized gold nanowire. This can be done by measuring the propagation length of the nanowire before and after moving a certain length. Electron diffraction pattern on different part of the wire would also helpful to understand if the single crystallinity is preserved.*

Reply to Comment: Thank you very much for the positive response to our work. We fully agree with this suggestion, and it is very helpful for improving our work. To address this issue, we experimentally tested an Au nanowire with a width of 170 nm, a thickness of 120 nm, and a length of ~6.3 μm , which were driven to move a distance of 3.5 μm by 1,064-nm light power of about 6.0 μW on a microfiber ($D_{\text{fiber}} = 1.5 \mu\text{m}$). The changes in nanowire lengths were characterized using a

high-resolution atomic force microscope (AFM, Asylum Research Cypher Oxford Instruments). The AFM scanning results in Figures (a) and (b) below show that the nanowire has no detectable change in the length before and after being driven by the nanosecond pulsed laser. In addition, we further used high-resolution transmission electron microscopy (TEM) to investigate the crystal behavior of as-driven nanowires. By removing a typical as-driven nanowire from a microfiber to a carbon film of copper mesh, Figure (c) below shows its electron diffraction patterns at the front, middle and back of the nanowire. From their regular lattice diffraction mottling, it can be seen that the nanowire has a single crystal behavior. These above results clearly indicate that the single crystallinity of Au nanowires is preserved and there is no recrystallization phenomenon in Au nanowires. In the revised manuscript, we have added these results as a new Figure in the Supplementary Information and related discussions in the main text.

Reviewer #2

In the paper entitled 'Plasmon-driven nanowire actuators for on-chip manipulation' the authors manipulate plasmonic gold nanowires with dielectric waveguides by applying laser pulse in the waveguides. When applying 532 nm light, the wire will be withdraw to the light source direction and when applying 1064 nm light, the wire will be driven to the light propagating direction because the plasmon induced surface acoustic wave make the wire crawl like a earthworm. Even if the wires have crossing angle with the fiber, the optical force will make the wire self-parallel parking through the fiber. With this technique, the authors demonstrated that they can manipulate the gold wires through U, C shaped fibers and transfer a wire to another fiber. This technique has potential use on the plasmonic wire based circuit. The technique is quite advanced and practical and the fiber induced surface acoustic wave here is very interesting. I think this work will attract the interest of broad readers and will bring impact in the future. So I recommend it for publication in nat. comm. However, there are some critical points need to be addressed (the importance is randomly listed).

Comment (1): *In the paper, the authors said that there was no detectable nanowire movement observed when continuous-wave lasers or femtosecond pulsed lasers were used, even though the laser power was high enough to damage the nanowires, indicating that the optical force induced by the photon momentum can also be excluded. So I conclude that the pulse width, repeating*

frequency are very important parameters for the wire driven technique. However, the authors didn't give any focus on the pulse width. it is only mentioned in the method part for 1064 nm light. For the doubled frequency 532 nm, if the width changed a lot? The authors should give proper theoretical or experimental prove together with the discussion in the main text because the acoustic wave of plasmonic wire should be considered (the decay time of a 100 nm particle is about 100 ps). For the repetition frequency, it is easier to understand that the wire moving speed should be linearly related to it. But the repetition should come earlier in the main text.

Reply to Comment (1): Before we address individual comments, we first would like to thank the reviewer for the positive response to our work. Your comments are very insightful and are very helpful to improve our manuscript.

Experimentally, we used other laser sources to investigate the nanowire movement behaviors. We used femtosecond pulsed laser from a commercial Ti:sapphire laser (Spectra Physics) producing 810-nm-wavelength with pulse duration of ~70 fs and a constant repetition rate of 80 MHz. We also used a 1,064 nm wavelength nanosecond pulsed laser from a solid state Q-switched laser (Changchun New Industries Optoelectronics Tech. Co., Ltd.), with pulse duration of 8.5 ns and tunable repetition rates from 1 Hz to 4 kHz. The used 532 nm laser pulses were generated from the 1,064-nm nanosecond laser by using a frequency-doubling KTP crystal, with pulse width of 5.6 ns and also tunable repetition rates. Recently, we used a broadband supercontinuum source (SuperK Compact, NKT photonics) with master source pulse width of ~1 ns and a constant repetition rate of 24 kHz, also used a 1035-nm-wavelength pulsed fiber laser with tunable pulse duration from ~600 fs to 6 ps and a repetition rate of 25 kHz (FemtoYL-50, Wuhan Yangtze Soton Laser Co.,Ltd.).

In all, obvious continuous and controllable movements were observed by using the nanosecond lasers (8.5 ns at 1,064 nm and 5.6 ns at 532 nm) and the supercontinuum source (1 ns). But, although picosecond (pulse duration: ~6 ps) and femtosecond pulses (pulse duration: ~600 fs) could drive the nanowires to move a few micrometers, the nanowires quickly stop moving. Then even if the laser power was increased until the nanowires were damaged, the nanowires still could not continue to move. AFM scans reveal that the surface roughness of as-driven nanowires has increased a lot, suggesting that the nanowire lattices were damaged by the pulsed lasers and there were a recrystallization or reconstruction effect on the nanowire surface.

For the theoretical calculations of the metal nanowire versus pulse duration, Comment (2), (3), and (4) also refer to this issue. We think the pulsed laser with duration around the nanosecond are effective sources that can continuously and controllably drive the metal nanowire along the microfiber, and the detailed responses can be found there. In the revised manuscript, we have added more discussions about the issue of pulse duration in the main text, and also added a new Supplementary Movie of the nanowire movement driven by the 664-nm long-pass filtered supercontinuum source.

For the repetition rates, we fully agree with that it is easy to understand that the moving speeds of the nanowires are linearly dependent on each pulse energy. In the revised manuscript, we have added some discussions about the relation between the repetition and the moving speed in the earlier part of "Plasmon-driven nanowire actuators." in the main text.

Comment (2): *For the movement detail, it is not explained quite clear in the paper. There have*

been many papers on the plasmon decayed lattice oscillation researching. For example, for a about 100 nm particle, usually the plasmons will decay in about 10s fs and transfer the energy to the lattices in about 1 ps time. Then there is acoustic wave oscillation and in about 100 ps, the oscillation will arrive at the balance. In this work, consider the length of the wire, the plasmons have much faster speed than the acoustic speed on the wire, and the lattice oscillation should happen almost the same time in the whole wire. But from the supporting information I see that the waves (from the main text I think it is the lattice oscillation waves) propagate to the end in about tens of nano seconds and the oscillation keeps over 100 ns. So what really happens here is not clearly addressed in the paper.

Reply to Comment (2): We thank the reviewer for pointing out this issue. There have been many papers on the plasmon decayed lattice oscillation researching, and typical references are listed below:

- [1] M. L. Brongersma, N. J. Halas, and P. Nordlander. Plasmon-induced hot carrier science and technology. *Nature Nanotechnol.* **10**, 25 (2015).
- [2] S. Link, C. Burda, B. Nikoobakht, and M. A. El-Sayed, Laser-induced shape changes of colloidal gold nanorods using femtosecond and nanosecond laser pulses. *J. Phys. Chem. B* **104**, 6152 (2000).
- [3] M. Hu, and G. V. Hartland. Heat dissipation for Au particles in aqueous solution: relaxation time versus size. *J. Phys. Chem. B* **106**, 7029 (2002).

For nanorods and nanoparticles used in these references, the dimensions are usually below 100 nm, which is much smaller than the wavelength scales of the light source used. But in our work, the used metal nanowires have the lengths of several or more micrometers, which are much larger than the wavelength scales of the light source. The heat dissipation process caused by surface scattering in nanorods and nanoparticles is faster than the thermal conduction process. But the in our nanowires with lengths on the order of micrometers, the temperature distribution (heat dissipation process) along the nanowires will depend on the thermal conduction process of the metal. Experimentally, we used pulsed lasers from nanoseconds to femtoseconds to drive the metal nanowires. However, for shorter nanowires with lengths less than $\sim 1.5 \mu\text{m}$, no detectable movements were observed. Therefore, we think our laser driven mechanism is a process that requires a meter length scale and a nanosecond timescale, and we need to consider the propagations of SPPs, heat, and surface acoustic waves along the metal nanowires, which are very different from those in nanorods and nanoparticles.

The interaction of light with metal can be typically divided into 3 processes. Upon excitation of SPPs at metal surfaces, the timescale for the process of electrons absorbing photons to generate hot electrons is on the order of 100 fs, and the hot carrier distribution is highly non-thermal. Subsequently, the hot electrons will redistribute their energy by electron–electron scattering processes (heating up the lattice) until reaching the thermal equilibration with the lattice, which is on a timescale ranging from ~ 100 fs to tens of ps. Finally, heat is transferred to the surroundings of the metallic structure via conduction, convection and radiation, which is on a longer timescale ranging from ~ 100 ps to ~ 10 ns or more depending on the material sizes and the thermal conduction properties of the environment.

In our work here, the motion mechanism is based on the synergistically working of expansion, friction, and contraction. The process of heat induced lattice expansion is an essential initial step for the nanowire movement. From the experimental results in our Reply to Comment (1), we think

the nanosecond pulsed lasers are effective sources that can continuously and controllably drive the metal nanowire along the microfiber. For theoretical calculations of pulse duration, Comment (3) also refers to this issue, and the detailed responses can be found there.

Due to the relative low propagation loss of SPP at 1,064 nm, the SPPs do not dissipate very fast, and will form standing waves along the nanowire–microfiber interface. The formation of SPP standing waves can be seen in our Response to Comment (6). The SPP standing waves will induce the heat power density spatially distributed like the standing waves, and the thermal energy is highly concentrated around the nanowire frontend. Then the generated thermal energy causes the lattices of Au nanowires to expand suddenly. This sudden expansion will propagate from the nanowire frontend to other areas in the form of mechanical oscillation (surface acoustic waves), at a speed of sound. By fitting the attenuation part of the time-dependent axial expansion curve of the nanowire frontend (see Figure 3e in the main text), a time constant of about 60 ns is obtained. At the same time, a small fraction of the heat is lost to the air or to the microfiber through heat conduction and convection, and most of the thermal energy at the nanowire frontend is transferred to the entire nanowire through thermal conduction until the heat distribution along the entire nanowire is balanced. By fitting the attenuation part of the time-dependent transient temperature curve of the nanowire frontend (see Figure 3f in the main text), a time constant of about 12 ns is obtained.

Therefore, due to the long length of nanowires and the propagation of surface acoustic waves, the equilibrium time required for thermal conduction and mechanical oscillation is much longer than that of heat dissipation in nanoparticles and nanorods. In the revised manuscript, to make the mechanism more clearly, we have added more discussions and references about this issue in the main text.

Comment (3): *Continue with the above comment, what happens if changing the pulse duration time? Maybe the authors don' have the instrument to sweep this parameter, but the authors have already done it with fs lasers, so I think at least for ps laser it should be included here for the mechanism.*

Reply to Comment (3): We thank the reviewer for this comment. Experimentally, obvious continuous and controllable movements were observed by using the nanosecond lasers (8.5 ns at 1,064 nm and 5.6 ns at 532 nm) and the supercontinuum source (1 ns). But, although picosecond (pulse duration: ~6 ps) and femtosecond pulses (pulse duration: ~600 fs) could drive the nanowires to move a few micrometers, the nanowires quickly stop moving. As our Reply to Comment (2), we think the pulsed laser with duration around the nanosecond are effective sources that can continuously and controllably drive the metal nanowire along the microfiber.

Because our mechanism is based on SPP induced heating effect, we investigate the influence of the pulse duration on the nanowire movement by theoretical calculations of the pulsed laser induced transient temperature in the nanowires. We use the eq.S8 in the Supplementary Information. The Gaussian temporal and spatial distribution is used to simulate the heat power density distribution, as follow:

$$Q_d(x, y, z, t) = P_0 \cdot Am \cdot \frac{M}{\sqrt{\pi\tau}} \exp\left[-\frac{(t-t_0)^2}{\tau^2}\right] \exp\left[-\frac{(x-x_0)^2}{s_x^2} - \frac{(y-y_0)^2}{s_y^2} - \frac{(z-z_0)^2}{s_z^2}\right]$$

The dependence between the coefficient (the time constant of the light pulse) τ in the temporal Gaussian distribution and the pulse duration t_p can be described as:

$$\tau = \frac{t_p}{2\sqrt{\ln 2}}$$

By integrating the heat power density with time, we can get the spatial distribution of the energy density:

$$E = P_0 \cdot A_m \cdot M \cdot \exp\left[-\frac{(x-x_0)^2}{s_x^2} - \frac{(y-y_0)^2}{s_y^2} - \frac{(z-z_0)^2}{s_z^2}\right],$$

where M is the time period of the pulse, A_m is the normalized power density from the FDTD simulation, and t_0 is the time delay of the pulse peak after the pulse is triggered. According to the energy density, as the pulse width is adjusted with a constant repetition rate, the energy of a single pulse remains unchanged, but the instantaneous power within the pulse duration will increase dramatically. In the numerical simulation, the pulse duration and the input power should be changed at the same time; otherwise the instantaneous heating power density may be much higher than the melting point of gold ($\sim 1,064$ °C). Batch adjustments are made during the simulation, and we set

$$P = P_0 \cdot \sqrt{\frac{t_{pt}}{t_p}}$$

where t_{pt} is the pulse duration adjusted, t_p corresponds to the value used in our work, and P is the corresponding power to t_{pt} . The pulse duration used in our paper (t_p) is 8.5 ns, the time constant (τ) is 5.0 ns, and the input power (P_0) is 0.347 mW. Then we run new simulations for the 6- μm -length nanowire with other parameters unchanged. By maintaining the time delay of the pulse peaks, the simulation results with different pulse duration (10 ns, 6 ns, 3 ns and 1 ns) can be obtained as follow:

We can see that as the pulse duration decreased, the absorption time in the nanowire is shortened accordingly, and the time to reach the maximum temperature peak will also be shortened. The calculated forward distances of the nanowire are 0.75 nm, 1.05 nm, 1.30 nm, and 1.05 nm, for the pulse duration of 10 ns, 6 ns, 3 ns, and 1 ns, respectively. These results suggest that the pulsed laser with durations around the nanosecond can induce moderate temperature increasing to drive the metal nanowire effectively.

For picosecond pulses, we used the heat conduction model (the eq.S8 in the supplementary Information) to simulate the pulses (using shorter time steps and smaller time delays of the pulse peak t_0) and compared the results with that of the 1 ns pulse. For short pulses less than 0.5 ns, the nanowire has obvious unbalanced spatial-temporal temperature response between the side and the center of the frontend. In the figure below, the dashed and solid lines show the calculated time-dependent transient temperature change at the side and center point respectively. At the very beginning of the light absorption, the heat power density at the nanowire bottom shows an unbalanced distribution for these short pulses, in which the temperature difference between the center and side points is very large. For example, the temperature differences between the center and side points at the nanowire bottom are 760 K, 1170 K and 590 K for time durations of 6 ps, 50 ps and 0.2 ns respectively (Fig.S16), but this difference in Fig.S11 is much smaller by using the nanosecond pulses, only 7 K. Such high temperature difference can not be ignored. We think that in such an ultra-short time scale (less than 0.2 ns), the heat transfer from the side point to the center point could not complete, and the highly concentrated heat at the side point may damage the nanowire lattice. AFM scans reveal that the surface roughness of as-driven nanowires has increased a lot, suggesting a recrystallization or reconstruction effect on the nanowire surface. For femtosecond pulses, the peak power of the pulses is much higher and will induce more unbalanced spatio-temporal temperature responses. Experimentally the phenomena of driving nanowires with femtosecond pulses (600 fs) and picosecond pulses (6 ps) were very similar.

In the revised manuscript, we emphasized that using nanosecond pulsed lasers can obtain **continuous and controllable** manipulation of single metal nanowires, and we also added some discussions about the pulse duration in the main text and the simulations in the Supplementary Note 3.

Comment (4): Consider above comment, in 1 ns pulse duration, the wire should be already thermalized balantly. If not, what is really the mechanism here.

Reply to Comment (4): As our reply to Comment (2), due to the long length of nanowires and the relative low propagation loss of SPP at 1,064 nm, the equilibrium time required for mechanical oscillation and thermal conduction is much longer than that of heat dissipation in nanoparticles and nanorods. The SPPs excited by the 1,064 nm laser does not dissipate very fast, and will form

the standing waves along the nanowire–microfiber interface. Upon the heat-induced sudden expansion at the nanowire frontend, most of the thermal energy is transferred to the entire nanowire through thermal conduction until the heat distribution along the entire nanowire is balanced. The time constant of this heat distribution process is about 12 ns (see the time-dependent transient temperature curve in Figure 3f in the main text).

Comment (5): Consider above comment, for difference length of wires, the movement speed should be different because the heating pulse takes more time to get the balance. The authors have done experiments for different length of wires, but didn't give any statistics or comment on it.

Reply to Comment (5): It is a very insightful comment. Experimentally, we have investigated the dependence of the nanowire lengths on the moving speeds by testing tens of Au nanowires. The nanowires used were placed on a same microfiber with a diameter of 2.3 μm , and the input power of the 1,064-nm nanosecond laser was maintained at $\sim 0.8 \mu\text{W}$. The results of the experimental statistics are shown in the Figure (a) below. We can see that for shorter nanowires with lengths less than $\sim 1.5 \mu\text{m}$, no detectable movements were observed. From the simulation in the Figure (b) below, we can see that under the same optical input power, the induced heat power density in a 1- μm -length nanowire is one order of magnitude lower than those of other two nanowires with lengths of 2 μm and 6 μm . The difference in the heat power density between the frontend and backend is also much smaller than those of other two nanowires. These unfavorable factors cause the shorter nanowires (lengths less than $\sim 1.5 \mu\text{m}$) unable to overcome the adhesion force and move.

For nanowires with lengths around $\sim 3 \mu\text{m}$, the maximum moving speeds were usually observed. But for much longer nanowires, their moving speeds were also decreased, which can be attributed to the increased adhesion force to the microfiber surface as the lengths increase. Under other laser power conditions, we also observed a similar behavior of the relationship between the nanowire lengths and the moving speeds.

In addition to the length, the moving speed of the nanowires on a microfiber also depends on other factors, such as the cross-sectional sizes of the nanowires, which have been referred in our Reply to Comment (14). In the revised manuscript, we have added some discussions about the nanowire speeds in the main text, and also added the figure below in the Supplementary Information.

Comment (6): Continue with the above comment, for the simulations, in FDTD method only pulse is used to excite samples and different wavelength post results are got from fourier transformation. So in 1 ns duration, the light has already oscillated for about 10^5 times, the heat effect should be an averaged value. For finite wire, there is standing wave, so the heating can be wave like distribution. Considering the energy dissipating, the heating energy in one end should be large than the other end. But if the SPPs dissipate along the wire very fast, there will be not standing wave. How did the authors get the thermal waves in the paper (especially Figure 2e)? I didn't find it in the methods part.

Reply to Comment (6): We thank the reviewer for this comment. Due to the relative low propagation loss of SPPs at the 1,064 nm wavelength, the SPPs do not dissipate very fast, and form the standing wave along the nanowire–microfiber interface. Comment (2) and (9) both refer to the SPP standing waves, so here we answer this issue together.

To clearly explain this issue, we used an FDTD method to simulate the interaction dynamic process between the microfiber and the nanowire. When guided light in the microfiber comes into contact with the nanowire backend, the HE_{11}^y mode of the 1,064-nm light immediately starts to excite the SPP around the nanowire-microfiber interface. Due to the relative low propagation loss of SPPs at 1,064 nm, the guided light in the microfiber and the excited SPPs propagate forward synchronously, and simultaneously the guided light excites new SPPs, which leads to the continuous increasing of the SPP energy along the nanowire bottom. When reaching the nanowire frontend (with respect to the guided light direction), the accumulated forward propagating SPPs will be reflected by the nanowire frontend and propagate backward, thus the forward and the backward SPPs interfere to form a standing wave, as shown in the figure below. Due to the existence of propagating loss, the intensity of the SPP standing wave reaches the maximum at the nanowire frontend, and quickly decays exponentially along the nanowire to the backend. In addition, it is known that the standing wave due to the overlapping can increase the maximum of light intensity by near 4 times, which will further enhance the heating effect along the nanowire bottom and be helpful to drive the nanowire. Moreover, due to the forming of the standing wave, the E field excited by the HE_{11}^y mode on the right side (frontend) of the nanowire is stronger than that on the left side (backend).

In the above mentioned FDTD simulations, the index of Au and SiO_2 are set to be $0.106 + 6.82i$ and 1.45 at 1,064 nm, respectively [D. R. Lide ed. *CRC Handbook of Chemistry and Physics*. 85th Edition, CRC Press, 2004]. We use a grid size (dx , 20 nm; dy , 10 nm; dz , 20 nm) for the Au nanowire-silica microfiber structure, and set the gap between the Au nanowire and microfiber as 1 nm. The propagation loss of the Au nanowire at 1,064 nm is calculated as $0.1 \text{ dB } \mu\text{m}^{-1}$ from the formula below [W. L. Barnes, A. Dereux, and T. W. Ebbesen, Surface plasmon subwavelength optics. *Nature* **424**, 824 (2003)]:

$$k_{spp} = k_0 \sqrt{\frac{\epsilon_m + \epsilon_d}{\epsilon_m \epsilon_d}},$$

where k_{spp} is the wavenumber (propagation constant) of the SPP, k_0 is the wavenumber of electromagnetic waves in vacuum, ϵ_m and ϵ_d is the relative permittivity of the gold and dielectrics, respectively.

The electric field distribution in the Au nanowire can be extracted from the results of FDTD simulations, and thus we can obtain the heat power volume density $Q_d(r, \lambda)$ (W/m^3) from the formula below:

$$Q_d(r, \lambda) = \frac{1}{2} \epsilon_0 \omega \text{Im}(\epsilon_r) |E(r, \lambda)|^2,$$

where ϵ_0 is the vacuum permittivity; ω is the frequency of light source; $\text{Im}(\epsilon_r)$ is the imaginary part of the relative permittivity of the gold. $E(r, \lambda)$ is the electric field intensity (V/m) with a specific wavelength in the Au nanowire.

In the revised manuscript, we have added more discussions about this issue in the main text, and also provide the simulated figure and movie as new supplementary materials.

Comment (7): *For the simulations, the authors didn't give the pulse duration information. If 1 ns is used here, then the covered wavelength range should be very limited. If it is available for the simulations.*

Reply to Comment (7): Thank you for reminding this point. Experimentally the measured duration of the used 1,064-nm pulsed laser is about 8.5 ns, so in our simulation, we also use the 8.5 ns as the pulse duration. In the revised manuscript, we have added this information in the numerical simulations in the main text and in the Supplementary Note.

Comment (8): *In Figure S4, the authors gives the E field distributions of the wire-fiber. From the figure one can see that the wire has little influence for the fiber. So why is the mode distribution an elliptic area but not a circular area? and From Figure S4b right column I see that the boundary is almost flat in the whole fiber no matter if there is wire or not.*

Reply to Comment (8): Thank you for pointing out this issue that may have caused misunderstanding. In fact, the mode distributions inside the microfiber in our previous manuscript (Figure S4 (a) and (b)) have a near circular area, but when we processed the simulated figures, we accidentally changed the aspect ratio of the figures, which made the figures look elliptical. Figure 2a in the main text also has a slight deformation. In the revised manuscript we have correct this issue.

When the nanowire is excited by HE_{11}^y and HE_{11}^x mode of guided 532-nm light in the microfiber, SPPs are difficult to be excited due to strong material absorption. But we can still observe little fluctuation at the microfiber boundary beneath the nanowire for the 532-nm HE_{11}^y mode, as denoted in the figure below.

Comment (9): In figure 2, why the E field on the right side of the wire is stronger should be explained in the text.

Reply to Comment (9): In Figure 2, we show the calculated distributions of electric fields and heat power densities in Au nanowires under the excitation of 1,064 nm light. The stronger E field excited by the HE_{11}^y mode on the right side (frontend) of the nanowire than that on the left side (backend) can be attributed to the **SPP standing wave** formed along the nanowire–microfiber interface. Due to the existence of propagating loss, the intensity of the SPP standing wave reaches the maximum at the nanowire frontend, and gradually attenuates along the nanowire to the backend. The details can be seen in our Reply to Comment (6). In the revised manuscript, we have added more discussions to stress on the SPP standing wave in the main text.

Comment (10): Is there any gradient force here in Figure 2?

Reply to Comment (10): Figure 2 shows the calculated distributions of electric fields and heat power densities in Au nanowires. Here we assume that the diameters of silica microfibers are uniform. Experimentally the silica microfibers were fabricated by using a simple flame-heated method, which have high uniform diameters [F. Liao, et. al, Enhancing monolayer photoluminescence on optical micro/nanofibers for low-threshold lasing. *Sci. Adv.* **5**, eaax7398 (2019)]. The diameter variations can be less than 10 nm over a 1-mm length [N. Yao, et al, Ultra-Long subwavelength micro/nanofibers with low loss. *IEEE Photon. Technol. Lett.* **32**, 1069 (2020)]. So the guided light intensity in the microfibers can be considered uniform along the length direction.

In addition, we also used an FDTD method to calculate the optical gradient force in an Au nanowire excited by 1,064-nm light, in which the size of each grid in the x , y , and z directions is $20 \text{ nm} \times 10 \text{ nm} \times 20 \text{ nm}$ respectively. The Au nanowire has a thickness of 100 nm, a width of 200 nm, and a length of $6.0 \text{ }\mu\text{m}$ and the microfiber has a D_{fiber} of $1.8 \text{ }\mu\text{m}$. The light force density

($F_{density}$) can be calculated from the equation below:

$$F_{density} = \frac{1}{2} \left[(P \cdot \nabla) E^* + \frac{\partial P}{\partial t} \times \mu_0 H^* \right],$$

where E and H are the usual electric and magnetic fields, P is the polarization density, μ_0 is the permeability of free space. The relevant calculation method can be found in our published references below:

- [1] H. Yu, et. al, “Longitudinal Lorentz force on a subwavelength-diameter optical fiber.” *Phys. Rev. A* **83**, 053830 (2011);
- [2] Y. Zhang, et. al, “Theoretical analysis of optical force density distribution inside subwavelength-diameter optical fibers.” *Chin. Phys. B.* **27**, 104210 (2018).

The calculated linear density distributions of light force along the nanowire-microfiber interface is shown in the figure below, in which the generated light force exhibit a gradient change along the nanowire. By integrating the force density along the nanowire-microfiber interface, the total light force is about 2.1×10^{-23} N, which is much smaller than the adhesion force of nanowire to microfiber in air environments (a level of $\sim \mu\text{N}$). Therefore, in our work, the gradient force can be negligible.

Comment (11): *In the paper the authors didn't give the mechanism of the pulling back effect for 532nm. Even if the title of the paper is 'plasmon driven', the pulling back effect for 532 nm should be explained in the paper. Otherwise the work is not complete.*

Reply to Comment (11): In our previous manuscript, the mechanism of the pulling back effect for 532nm has been introduced in the main text with several sentences. The heating focus of the 532-nm light appears at the opposite end to that of the 1,064-nm light, so the nanowire moves in the opposite direction with the 532-nm light. In addition, in Supplementary Information we also provided the simulated electric field distributions of the nanowire-microfibre system excited by the 532-nm light. To make the mechanism for the 532-nm light more clear, we have added more sentences to emphasize this issue in the main text, and details can be found in the revised manuscript (also see sentences below).

“From the motion mechanism proposed above, we can also see that the heating positions formed at the frontend or backend of the nanowire will determine the movement direction. For shorter wavelengths such as the 532 nm wavelength (Fig. 1c), due to the strong

absorption of Au material to light, no obvious SPP is excited and propagates in the nanowire, thus no standing wave will be formed around the nanowire bottom (Supplementary Movie 2). Only the direct thermal absorption of the HE_{11}^y mode light (Supplementary Fig. 11) generates a heating source around the nanowire backend, and thus driving the nanowire to the opposite direction of the light. In addition, it is also noticed that the maximum magnitude of the heat power density induced by the 532 nm light is an order of magnitude lower than that induced by the 1,064 nm light, which is attributed to the absence of the SPP standing wave. So experimentally, much higher power of the 532 nm nanosecond laser is usually needed to drive the nanowires, but which may bring damage to the nanowires.”

Comment (12): *Indeed this method can manipulate the wire with very accurate steps along the wire, but the position accuracy also depends on the fiber position. And the fiber position is still controlled through the scanning stage or other mechanical methods. So in the introduction and conclusion part, the authors said for accurately control the wire for optical chip is a little bit overselling and should be modified.*

Reply to Comment (12): Thank you for pointing out this issue. In the revised manuscript, we have changed the mentioned expressions in the introduction and conclusion parts to avoid overselling.

Comment (13): *In Figure s5a, why is the heating distribution is so asymmetric?*

Reply to Comment (13): In this Supplementary Figure, we provide the heat power density distributions on the bottom surface of an Au nanowire, excited by the guided HE_{11}^y mode at a 457 nm wavelength. The total electric fields are composed of the excited SPPs and the evanescent field from the microfiber and its scattering. Since the SPP is hardly excited at a 457 nm wavelength, the total electric field is mainly determined by the evanescent field from the microfiber and its scattering. The Figure (a) below shows the electric field distribution on the bottom surface of an Au nanowire. We can see that the electric field distribution is very asymmetric, and the intensity is also smaller than that in the microfiber. Since the heat power density is proportional to the square of the electric field, the resulting heating distribution also looks asymmetric (Figure (b)).

Comment (14): *The effect of the diameter of the wire should be discussed.*

Reply to Comment (14): Thank you for pointing out this issue. We experimentally tested two Au nanowires with different cross-sections but approximately the same lengths ($L_{NW} = 4.8 \mu\text{m}$), which were driven by the same 1,064-nm light power ($3.1 \mu\text{W}$) and on the same microfiber ($D_{\text{fiber}} = 2.1 \mu\text{m}$). The used laser repetition rate was 4 kHz. It shows that the moving speed ($10.9 \mu\text{m s}^{-1}$) of the thick nanowire (**a**: $360 \text{ nm} \times 150 \text{ nm}$ in width and thickness) is lower than that ($18.9 \mu\text{m s}^{-1}$) of the thin nanowire (**b**: $160 \text{ nm} \times 110 \text{ nm}$ in width and thickness). We also calculated the distributions of heat power densities in the two nanowires, as shown in Figure (c) below. It can be observed that the maximum heat densities in two nanowires are on the same order of magnitude. The larger moving speed of the thinner nanowire can be attributed to its smaller adhesion force to the microfiber surface. In the revised manuscript, we have added these results as a new figure in the Supplementary Information and also some discussions in the main text.

List of changes in main text of the revised manuscript:

(1). Throughout the article:

- The word “microfiber” is changed to “microfibre”;
- The word “microfibers” is changed to “microfibres”;
- The word “fiber” is changed to “fibre”;
- The word “fibers” is changed to “fibres”;
- The word “1064” is changed to “1,064”;

(2). In page 1:

- Line 24, the word “that” is deleted;
- Line 26, the words “plasmon-driven” are added;
- Line 27, the words “by plasmon driving” are deleted;
- Line 28, the word “enhance” is changed to “enhances”;
- Line 32, the word “eventually” is deleted.

(3). In page 3:

Line 4, the sentences “Here, we propose an earthworm-like peristaltic crawling motion mechanism based on the synergistically working of expansion, friction, and contraction, and experimentally demonstrate manipulation of single metal nanowires in non-liquid environments by plasmon driving, with advantages of sub-nanometer positioning accuracy, low actuation power, and self-parallel parking. Capitalizing on this approach, we further perform on-chip nanowire manipulations of hybrid photonic-plasmonic circuits including transporting, positioning, coupling, and sorting, to demonstrate the advantages of on-situ operation, high selectivity, and great versatility.” are changed to “Here, we propose an earthworm-like peristaltic crawling motion mechanism based on the synergistically working of expansion, friction, and contraction, and experimentally demonstrate continuous and controllable manipulation of single metal nanowires on fixed microfibrils in non-liquid environments by plasmon driving, with advantages of sub-nanometer positioning accuracy, low actuation power, and self-parallel parking. Capitalizing on this approach, we further perform on-chip manipulations of single nanowires on fixed microfibrils in hybrid photonic-plasmonic circuits including transporting, positioning, coupling, and sorting, to demonstrate the advantages of on-situ operation, high selectivity, and great versatility.”;

Line 16, the words “and fixed” are added;

Line 27, the sentences “In addition, as increasing the repetition rates of the pulsed lasers, the moving speeds of the nanowires also increased linearly. For nanowires with much larger cross-sections (Supplementary Fig. 3) or longer lengths (Supplementary Fig. 4), their moving speeds decreased compared with those with shorter lengths or smaller cross-sections, which can be attributed to the increased adhesion force to the microfibril surface. But for much shorter nanowires with lengths less than $\sim 1.5 \mu\text{m}$, no detectable movement was observed (Supplementary Fig. 4).” are added.

(4). In page 5:

Line 1, the words “In addition,” are deleted;

Line 1, the word “such” is changed to “Such”;

Line 2, the words “Supplementary Fig. 3” are changed to “Supplementary Fig. 5”;

Line 5, the sentences “In addition, no recrystallization phenomena in as-driven Au nanowires were observed by using a high-resolution atomic force microscope (AFM) and a transmission electron microscopy (Supplementary Fig. 6). Other laser sources (see Methods) were also used to investigate the nanowire movement behaviors. Similar movements were also observed by using a supercontinuum source with pulse duration of ~1 ns (Supplementary Movie 1).” are added;

Line 10, the sentences “were used, even though the laser power was high enough to damage the nanowires, indicating that the optical force induced by the photon momentum can also be excluded²⁸. Although 1,035-nm-wavelength picosecond (pulse duration: ~6 ps) and femtosecond pulses (pulse duration: ~600 fs) could drive the nanowires to move a few micrometers, the nanowires would quickly stop moving. Then even if the laser power was increased until the nanowires were damaged, the nanowires still could not continue to move. AFM scans reveal that the surface roughness of as-driven nanowires increased a lot (Supplementary Fig. 7), suggesting a recrystallization or reconstruction effect on the surface.” are added;

Line 17, the sentences “or femtosecond pulsed lasers were used, even though the laser power was high enough to damage the nanowires, indicating that the optical force induced by the photon momentum can also be excluded²⁸.” are deleted;

Line 18, the Fig. 2 is changed to a new one.

(5). In page 6:

Line 22, the words “Supplementary Fig. 4” are changed to “Supplementary Fig. 8”;

Line 26, the words “in longer wavelength ranges” are added;

Line 27, the words “Supplementary Fig. 5” are changed to “Supplementary Fig. 9”.

(6). In page 7:

Line 7, the words “guided in the microfibre” are added.

Line 8, the sentences “due to the relative low propagation loss of SPPs at 1,064 nm, the guided light propagates forward synchronously with the excited SPPs, and simultaneously excites more new SPPs, resulting in a continuous increase in the SPP energy along the nanowire bottom (Supplementary Fig. 10 and Supplementary Movie 2). When reaching the nanowire frontend (with respect to the guided light direction), the accumulated forward propagating SPPs will be reflected by the nanowire frontend and propagate backward, thus forming a standing wave at the nanowire-microfibre interface. Due to the existence of propagating loss, the intensity of the SPP standing wave reaches the maximum at the nanowire frontend and then decays toward the backend. The SPP standing wave will further enhance the heating effect at the nanowire frontend and” are added;

Line 16, the sentences “the heat power density gradually increases along the nanowire and reaches a maximum at the nanowire frontend (relative to the guided light direction). Such a gradient distribution” are deleted;

Line 17, the sentences “such a gradient distribution of the heat power density is beneficial for the nanowire movement as discussed below.” are added;

Line 18, the words “is beneficial for the nanowire movement.” are deleted;

Line 18, the sentences “Moreover, for shorter nanowires with lengths less than $\sim 1.5 \mu\text{m}$, the SPP excitation efficiency of the nanowire by the microfibre will decrease and the generated heat power density is much smaller than those in longer nanowires (Supplementary Fig. 4), causing shorter nanowires unable to overcome the adhesion force and move.” are added.

(7). In page 8:

Line 3, the words “Supplementary Movie 1” are changed to “Supplementary Movie 3”;

Line 18, the sentences “At the same time of the SAW propagation, most of the thermal energy at the nanowire frontend is transferred to the entire nanowire through thermal conduction until the heat distribution along the entire nanowire is balanced. ” are added;

Line 23, the sentences “By fitting the attenuation part of the response curve, a time constant of about 12 ns is obtained. Due to the long length of nanowires and the propagation of surface acoustic waves, the equilibrium time required for thermal conduction and mechanical oscillation is much longer than that of heat dissipation in nanoparticles and nanorods^{31–33}.” are added;

Line 28, the sentences “From the motion mechanism proposed above, the process of heat induced lattice expansion is an essential initial step for the nanowire movement, and in principle, as long as the lattice expansion can occur, our driving mechanism can be applied to other light sources to drive the nanowires. However, for picosecond (less than ~ 200 ps) and femtosecond pulses, their high peak energy can induce large temperature difference between the centre and side points of the nanowire frontend at the very beginning of the light absorption (Supplementary Note 3). In such a ultra-short time scale, the heat transfer from the side point to the centre point could not complete, and the highly concentrated heat at the side point may damage the nanowire lattice. So to obtain continuous and controllable manipulation, the pulsed lasers with duration around the nanosecond scale are the very effective sources to drive the metal nanowires.” are added.

(8). In page 9:

Line 8, the sentences “In addition, the heating positions formed at the frontend or backend of the nanowire will determine the movement direction. For shorter wavelengths such as the 532 nm wavelength (Fig. 1c), due to the strong absorption of Au material to light, no obvious SPP is excited and propagates in the nanowire, thus no standing wave will be formed around the nanowire bottom (Supplementary Movie 2). Only the direct thermal absorption to the HE_{11}^y mode light (Supplementary Fig. 11) generates a heating source around the nanowire backend, and thus the nanowire will be driven to the opposite direction of the light. It is also noticed that the maximum magnitude of the heat power density induced by the 532 nm light is an order of magnitude lower than that induced by the 1,064 nm light, which is attributed to the absence of the SPP standing wave. So experimentally, much higher power of the 532 nm nanosecond laser is usually needed to drive the nanowires” are added.;

Line 12, the sentences “The above motion mechanism can also be used to explain the movement behavior induced by 532-nm laser (Fig. 1c), but the heating source, directly from the thermal absorption of HE_{11}^y mode light (Supplementary Fig. 6), is concentrated

around the nanowire backend, thus inducing the opposite moving direction with that of the 1064-nm laser. In addition, for the 532-nm laser, the absence of high energy-confining strategy of SPPs usually needs high laser power to actuate the nanowires” are deleted; Line 13, the sentence “but which can damage the nanowire.” is changed to “which can bring damage to the nanowires.”.

(9). In page 10:

Line 11, the word “axial” is added;

Line 13, the word “bottom” is changed to “frontend”;

(10). In page 11:

Line 2, the words “Supplementary Movie 2” are changed to “Supplementary Movie 4”;

Line 12, the words “, which exhibits a linear relationship” are deleted;

Line 17, the words “Supplementary Fig. 7” are changed to “Supplementary Fig. 12”;

Line 21, the words “ (actuation average power typically exceeds tens of milliwatts) ^{24,31,32,} are changed to “(actuation average power typically exceeds tens of milliwatts) ^{24,34,35,}”.

(11). In page 12:

Line 8, the sentences “Dependence of the moving speed of the nanowire on the laser repetition rates. Error bars are the variance of moving speeds.” are changed to “Dependence of the moving speed of the nanowire on the laser repetition rates. Error bars are the variance of the moving speeds.” ;

Line 15, the sentences “We first demonstrate some basic on-chip nanowire manipulation such as transport, positioning, and orientation.” are changed to “We first demonstrate some basic on-chip manipulation of the Au nanowires on the fixed microfibres such as transport, positioning, and orientation. ”.

(12). In page 13:

Line 3, the words “Supplementary movie 3” are changed to “Supplementary Movie 5”;

Line 10, the word “fixed” is added;

Line 16, the word “fixed” is added;

Line 20, the words “evanescent force ^{12,33,} are changed to “evanescent force ^{12,36,}”.

(13). In page 14:

Line 13, the words “continuously and controllably” are added;

Line 14, the word “fixed” is added;

Line 16, the sentence “thereby greatly enhancing the heating effect of absorbed light.” is changed to “and form standing wave around the nanowire frontends, which enhances the heating effect of absorbed light and is beneficial for the nanowire movement.”.

(14). In page 15:

Line 4, the word “fixed” is added;

Line 5, the words “waveguides ^{13,14,34,} are changed to “waveguides ^{13,14,37,}”.

Line 9, the sentence “we may eventually realize full co-integration of various functionalized photonic components on single chips.” is changed to “we might realize co-integration of various functionalized photonic components on single chips.”;

Line 14, the word “method” is changed to “method^{38,39}”;

Line 15, the word “method” is changed to “method^{7,26}”;

Line 20, the sentence “For robust operation, microfibers were bound to an MgF₂ substrate using a low-index UV cured fluoropolymer (EFIRON PC-373; Luvantix Co. Ltd.)^{7,26}.” is changed to “For robust operation, microfibrils were bound fixed on the MgF₂ substrate using a low-index UV cured fluoropolymer (EFIRON PC-373; Luvantix Co. Ltd.)”;

Line 24, the words “Supplementary Fig. 8” are changed to “Supplementary Fig. 13”;

Line 26, the sentences “Experimentally, we used several laser sources to investigate the nanowire movement behaviors. The used 1,064 nm wavelength nanosecond pulsed laser was from a solid state Q-switched laser (Changchun New Industries Optoelectronics Tech. Co., Ltd.), with pulse duration of 8.5 ns and tunable repetition rates from 1 Hz to 4 kHz. The used 532 nm pulsed laser was generated from the 1064-nm nanosecond laser by using a frequency-doubling KTP crystal, with pulse width of 5.6 ns and also tunable repetition rates. The pulsed lasers were linearly polarized with TEM₀₀ transverse mode. The used supercontinuum source (SuperK Compact, NKT photonics) with master source pulse width of ~1 ns and a constant repetition rate of 24 kHz, was filtered by a the 664-nm long-pass filter to obtain light in the longer wavelength range. The used 1,035-nm-wavelength pulsed fibre laser has tunable pulse duration from ~400 fs to 6 ps and a repetition rate of 25 kHz (FemtoYL-50, Wuhan Yangtze Soton Laser Co.,Ltd.).” are added.

(15). In page 16:

Line 7, the sentences “Nanosecond pulsed lasers were used here as the actuation light sources, including a tunable 1,064-nm pulsed laser (pulse duration: 10 ns, repetition rate: 1 Hz to 4 kHz) and its 532-nm double frequency laser (Supplementary Fig. 9). The pulsed lasers were linearly polarized with TEM₀₀ transverse mode.” are deleted;

Line 10, the words “Supplementary Fig. 14” are added.

(16). In page 17:

Line 44, the reference “31. Brongersma, M. L., Halas, N. J. & Nordlander, P. Plasmon-induced hot carrier science and technology. *Nat. Nanotechnol.* 10, 25–34 (2015).” is added;

(17). In page 18:

Line 2, the reference “32. Link, S., Burda, C., Nikoobakht, B. & El-Sayed, M. A. Laser-induced shape changes of colloidal gold nanorods using femtosecond and nanosecond laser pulses. *J. Phys. Chem. B* 104, 6152–6163 (2000).” is added;

Line 5, the reference “33. Hu, M. & Hartland, G. V. Heat dissipation for Au particles in aqueous solution: relaxation time versus size. *J. Phys. Chem. B* 106, 7029–7033 (2002).” is added;

Line 7, change the number of reference into “34”;

Line 9, change the number of reference into “35”;

Line 11, change the number of reference into “36”;

Line 13, change the number of reference into “37”;

Line 15, the reference “38. Gu, Fu. et al. Single whispering-gallery-mode lasing in polymer bottle microresonators via spatial pump engineering. Light Sci. Appl. 6, e17061 (2017).” is added;

Line 17, the reference “39. Liao, F. et al. Enhancing monolayer photoluminescence on optical micro/nanofibers for low-threshold lasing. Sci. Adv. 5, eaax7398 (2019).” is added;

REVIEWERS' COMMENTS

Reviewer #1 (Remarks to the Author):

The authors have addressed my questions satisfactorily.

Reviewer #2 (Remarks to the Author):

I read through out the materials and I think it is acceptable for present form.

Author's Response to Reviewer #1

The authors have addressed my questions satisfactorily.

Reply to Comment: Thank you very much for the positive response to our work.

Author's Response to Reviewer #2

I read through out the materials and I think it is acceptable for present form.

Reply to Comment: Thank you very much for the positive response to our work.